

# Bacterial profile of bovine mastitis in Ethiopia: a systematic review and meta-analysis

Ephrem Toma Tora[1], Nahom Belay Bekele[2] and R. S. Suresh Kumar[3]

[1] Animal Sciences, Arba Minch University, Arba Minch, Southern Ethiopia, Ethiopia
[2] Livestock Research Office, Bonga Livestock Research Center, Bonga, Southern Ethiopia, Ethiopia
[3] Chemistry, Arba Minch University, Arba Minch, Southern Ethiopia, Ethiopia

## ABSTRACT

**Background:** Bovine mastitis is the commonest episode of infection in the dairy industry, which often occurs after damage of epithelial cells lining the teat duct. It is ranked as the second most important cause of milk production loss directly and a devastating disease with a higher incidence leading to the culling of dairy cows. Thus, this systematic review and meta-analysis is aimed to quantitatively estimate the current status of mastitis in general and bacterial mastitis particular in Ethiopia.

**Methods:** A literature search was carried from major databases and indexing services including PubMed, Google Scholar, Science Direct. Also, local institution repositories were searched to retrieve unpublished MSc and PhD theses. All studies were included addressing the prevalence of mastitis and bacterial isolates conducted in Ethiopia. Microsoft Excel was used to extract data and was imported to R Studio for the analyses. The random-effects model at a 95% confidence level was used for pooled estimates of outcomes. The degree of heterogeneity was computed by Higgins's $I^2$ statistics. Publication bias was checked by using the funnel plots of standard error augmented by Begg's and Egger's tests.

**Results:** A total of 46 studies with 15,780 cows were included in this study. All studies have collected 18,478 suspected samples for bacterial isolation. While pooled prevalence estimate of mastitis was 47.6%, the bacterial isolates pooled prevalence was 33.1%. The bacterial mastitis was 6.5% in cows infected by clinical patients and was 28.3% subclinical patients. The common isolates were *Staph aureus*, *Strep* species, *Staph epidermis*, *Escherchia coli*, *Corynebacterium bovis*, Coagulase Negative *Staphylococcus* and *Klebsiella* species. A univariate meta-regression analysis evidenced that the type of mastitis and management system was a possible source of heterogeneity (*P*-value = 0.001).

**Conclusion:** The pooled prevalence of bacterial mastitis in Ethiopian dairy cattle was high. The analysis showed bacterial pathogens like *Staphylococcus aureus*, *Staph epidermis*, *Strep* species and Coagulase Negative *Staphylococcus* are majorly accounted for bovine mastitis in Ethiopia. Therefore, the highly prevalent and commonly isolated pathogens cause contagious mastitis which require immediate attention by dairy producers to put under control by devising robust mastitis prevention and control interventions.

Corresponding author
Ephrem Toma Tora,
adech.tor@gmail.com

# INTRODUCTION

Ethiopia is greatly dependent on agriculture, particularly livestock production, which represents a huge national resource and forms an integral part of the agricultural production system and livelihood of the society (*FAO, 2019*). Ethiopia is believed to have the largest livestock population in Africa, with a total cattle population estimated to be about 65.3 million. Likewise, the cows represent 55.9% of the cattle population of the country and around 20.7% of the total cattle heads are milking cows. From the total female cattle population, 97.9% are local breeds. The cross and exotic breeds constitute 1.82% and 0.28%, respectively (*CSA, 2020*).

Milk produced from dairy cows provides an important dietary source for the majority of rural as well as considerable number of the urban and peri-urban population (*Muluye, Alemayehu & Gizaw, 2017*). Out of the total national milk production, between 85% and 89% is contributed from cattle, followed by goat, camel and sheep (*CSA, 2020*; *FAO, 2014a*). However, the amount of milk produced from those cows is by far below the national demand for milk and milk products in the country, due to various causes, out of which disease of the mammary glands known as mastitis is among the various factors contributing to reduced milk production (*Ahmed, Ehui & Assefa, 2004*).

Bovine mastitis, an inflammatory alteration of the parenchyma of mammary gland, results in significant morbidity and mortality of dairy cows. Mastitis is characterized by the complex and multi factorial agents, and its occurrence depends on variables related to the animal, environment and pathogen (*Muturi, 2020*). Epidemiologically, the causative agent can be categorized into contagious and environmental mastitis (*Abebe et al., 2016*; *Blowey & Edmondson, 2010*). Contagious pathogens are those an udders of infected cows serve as major reservoir, which include *Staphylococcus aureus, Streptococcus agalactiae, Mycoplasma* and *Corynebacterium bovis* (*Girma et al., 2012*; *Abebe et al., 2016*). On the other hand, environmental mastitis can be defined broadly as those intra-mammary infections caused by pathogens whose primary reservoir is the environment in which the cow lives. Environmental pathogens include *Escherchia coli, Klebsiella* species, *Streptococcus dysgalactiae, Streptococcus uberis*, and the majority of infections caused by these pathogens are clinical and of short duration (*Blowey & Edmondson, 2010*; *Seegers & Fourichon, 2003*).

Likewise, the presence or absence of clear clinical signs for mastitis can be either clinical or subclinical (*Seegers & Fourichon, 2003*). The clinical mastitis is manifested by sudden onset, alteration of milk composition and appearance, and the presence of the cardinal signs of inflammation in affected quarters of the udder and decrease of milk production (*Abebe et al., 2016*). The sub-clinical mastitis has no visible signs either on the udder or in the milk, but the milk production decreases and the somatic cell count increases; thus, it is more common and has serious impact (*Muturi, 2020*). Moreover, mastitis had been known to cause a great deal of loss or reduction of productivity as it causes financial loss as a result of the influence on the quality and quantity of milk yield, discarded milk following
antibiotic therapy, veterinary expense and culling mastitic cows at productive age (*Girma et al., 2012*; *Shiferaw & Telila, 2017*).

Over the years, a number of researchers have reported the prevalence of bovine mastitis in Ethiopia. Accordingly, the apparent prevalence of mastitis (*i.e.* without classifications) falls within the range from 9.12% to 88.9% (*Abebe et al., 2016*; *Shiferaw & Telila, 2017*; *Adane, Gizaw & Amde, 2017*; *Bedele et al., 2019*; *Dereje et al., 2018*; *Lakew, Fayera & Ali, 2019*; *Kitila & Kebede, 2021*; *Kumbe et al., 2020*; *Mitiku et al., 2017*; *Tezera & Ali, 2021*). According to the reports, it is the second ranked and most important cause of milk production loss directly and a devastating disease with a higher incidence leading to the culling of dairy cows (*Woods, 1987*). Despite the numerous reports on the widespread occurrence of mastitis in different parts of the country, a systematic review and pooled quantitative documentation of the status of bacterial mastitis has not reported so far. It is of paramount importance to consolidate the pooled prevalence estimates and develop an action plan to control and manage the disease that would help to reduce its prevalence and effects (*Ismael, 2018*).

The differences in etiology of mastitis and husbandry practices are yet to be systematically reviewed in Ethiopia. Also there was presently no quantitatively synthesized report of mastitis describing the spatial-temporal distribution. This systematic review and meta-analysis summarized available data on the etiology of bacterial mastitis published in the past 22 years, with the aim of improving current knowledge of bacterial mastitis, the types of mastitis registered and husbandry practices across regions of Ethiopia. Therefore, the systematic review and meta-analysis was aimed to (i) estimate pooled prevalence of bovine mastitis and (ii) identify the major bacterial causative agents which are relevant predictors that could be possibly accounted for heterogeneity in prevalence estimates between the recent reports.

## METHODS

This meta-analysis study was carried out based on the PRISMA guideline (Preferred Reporting Items for Systematic Reviews and Meta-Analyses) (*Moher et al., 2009*). The use of the PRISMA checklist entails the inclusion of relevant information in the analysis. The prime outcome of this study is to estimate the pooled prevalence of bovine mastitis in general and bacterial mastitis in particular in Ethiopia.

### Study area description

This systematic review and meta-analysis was conducted in Ethiopia, a country found in the horn of Africa located between 3°00′–15° 00′ N latitude and 32°30′–48° 00′ E longitude. Ethiopia has a land area of 1.04 million km$^2$ and a population of 116 million (*Lakew, Tolosa & Tigre, 2009*), the second most populous nation in Africa next to Nigeria (*FAO, 2014b*). Ethiopian climate is suitable for agriculture and it is also home for an estimated 60.4 million heads of cattle (*CSA, 2020*). Ethiopia has a diverse topography, which forms the basis for different agro-climatic zones. The area locates 2,300 m above sea level (m.a.s.l.) is considered highland. The highland region surrounded by a temperate

transition zone between 1,500 and 2,300 m.a.s.l., called as midland, while area with an altitude below 1,500 m.a.s.l. is classified as lowland (*Etana et al., 2020*).

*Literature search strategy*

The study involved a comprehensive search of literatures reporting bovine mastitis in Ethiopia that effected from February, 2021 to April, 2021. It was implemented using different key words in PubMed, Science Direct, and Google Scholar. Bulletin of Animal Health and Production in Africa and the Ethiopian Veterinary Journal were also searched for directly inaccessible online databases. Also, local institutional repositories were searched to retrieve unpublished MSc and PhD theses. The following Medical Subjects Headings (MeSH) terms were used in electronic search engine: "Mastitis" and "Ethiopia". Besides, the search processes were used including additional terms such as "Prevalence", "Bovine", "Bacteria", "Clinical", and "Subclinical". Selected studies were checked by cross-references of the collected studies.

## Selection of studies and data extraction

Articles that have studied the prevalence of bovine mastitis in Ethiopia were downloaded and added to the Mendeley reference manager. The inclusion and exclusion criteria were set for quality before starting review processes. Then, all studies selected through the search strategy were independently sorted out by the two authors (ET and NB).

The criteria for the selection of research articles, published reports and research thesis were centered on the prime objectives of this study, which are mentioned elsewhere. Accordingly, a study to be included in the meta-analysis had to fulfill the following eligibility criteria: (1) be published in English language, (2) be studied by cross-sectional design, (3) include California mastitis test (CMT) and/or clinical examination method, and etiology of bacterial mastitis (4) use a sample size of greater than 65. While extracting studies, titles and abstracts that were not found relevant to the outcomes of interest or did not fulfill the eligibility criteria were excluded.

The authors extracted individually important data related to study characteristics. The primary data extracted from the eligible studies were the year of study, type of mastitis, total number of observations, and the total positive samples. The explicit information about the major bacteria such as *Staphylococcus* species, *Streptococcus* species, Coliformis, *Corynebacterium*, *Mycoplasma* species and other major isolates were extracted based on the number of cases. Moreover, the first author's name, year of publication, year of study, study site, agro-ecological zone, and diagnosis methods were also extracted. All the extracted data were encoded in a Microsoft Excel (2010) spreadsheet. The details about the study screening strategy and exclusion reasons are presented in Fig. 1.

## Critical appraisal of studies

The critical appraisal of the internal and external validity of studies and to reduce the risk of biases was conducted according to the Joanna Briggs Institute's critical appraisal tool adapted for prevalence studies (*Munn et al., 2015*). The tool includes nine grade points, and each study was graded according to these tools by the study authors. The two authors
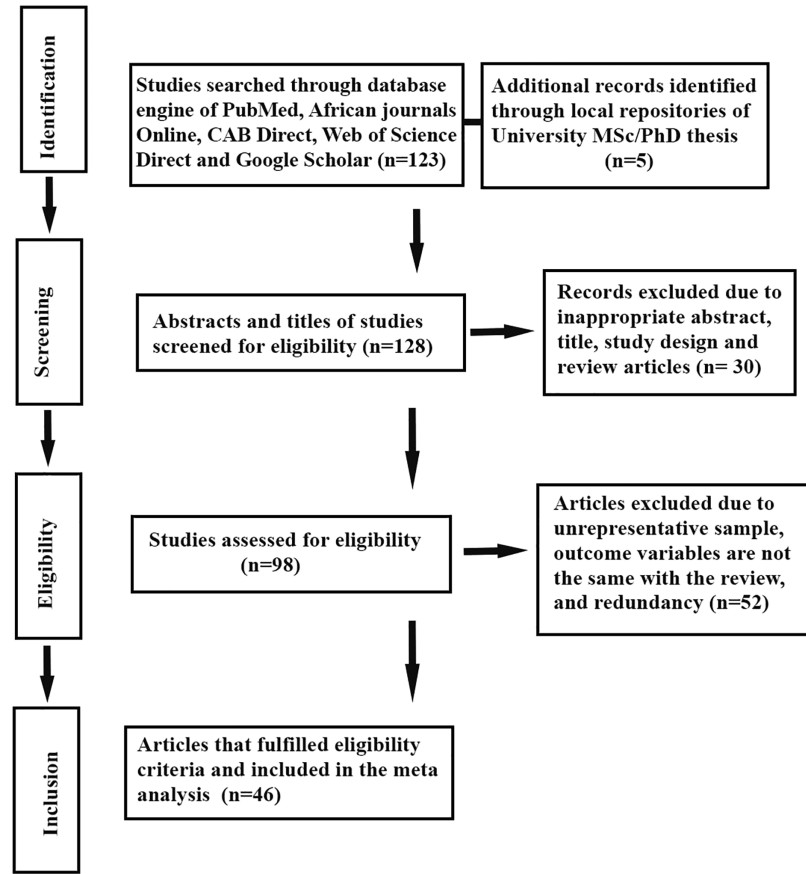

**Figure 1 Schematic representation of literature selection procedure adopted for the systematic review of bovine mastitis prevalence in Ethiopia.**

who selected the literature provided a score to each paper and selected those with greater and equal to five scores.

## Outcome measurements

The primary outcome of this study was to estimate the prevalence of bacterial mastitis in Ethiopia, screening for the major isolates involved in udder infection and the subgroup analysis of potential predictors.

## Literature bias assessment

Articles were evaluated to observe the inter-study bias across all collected literatures for quality assessment, because objectivity and consistency were a priority. Publication bias was assessed by visual examination of funnel plot, followed by the Egger's regression asymmetry test (*Egger et al., 1997*). Meta-regression was used to investigate factors that potentially contribute to the study heterogeneity. The Duval and Tweedie non-parametric 'fill and trim' linear random method were used to calculate the unbiased estimates (*Duval & Tweedie, 2000*).

## Statistical analysis

Meta-analysis was executed for the studies to estimate the pooled prevalence of mastitis and major pathogens known to cause bacterial mastitis (*Staph. aureus*, *Strep. dysaglactia*, *Strep. aglactia*, Coliformis, Coagulase negative S*taphylococcus*, *Corynebacterium species* and for other pathogenic bacteria). Analyses were also executed if data stratified by pathogen, type of mastitis, and production systems were presented from at least three studies.

Moreover, analyses were executed on the frequency of the pathogens in the agro-ecological zones (*i.e.* the high-, mid- and low land) to determine the impact that climatic condition may have. In computation of pooled estimate, two approaches were followed to combine the study results; the random-effects model was used for studies with moderate or high heterogeneity, and the fixed-effect model for those with low heterogeneity (*Egger et al., 1997*; *Wang, 2018*). The inverse-variance method was applied for the pooled summary measure estimation based on the Fixed-effect model (*Lakew, Tolosa & Tigre, 2009*) and the DerSimonian and Laird method for the random-effects models (*DerSimonian & Laird, 2015*). The Freeman-Tukey transformation was used to prevent the exclusion of study estimates that had 0 value (*Viechtbauer, 2010*). The Cochran's Q test was used to assess the level of heterogeneity. The Higgins $I^2$ test was used to quantify heterogeneity (*Higgins & Thompson, 2002*). Heterogeneity was categorized as low ($I^2$ of 0–40%), moderate (30–60%), substantial (50–90%), or high (75–100%) (*Higgins & Thompson, 2002*; *Wang, 2018*). The *P*-values were computed by comparing the statistic with the Chi-squared distribution with $k - 1$ degrees of freedom, and results with $P \leq 0.05$ were considered statistically significant heterogeneity (*Munn et al., 2015*). The GUI based R program embedded in R-Studio software (4.0.4 version) was used to analyze data (*R Core Team, 2020*).

# RESULTS

## Search results

This systematic review and meta-analysis were conducted per the PRISMA statement (*Moher et al., 2009*). A total of 128 studies were identified and sorted from various database sources. From the studies, 30 studies were excluded by manual tracing due to inappropriateness of abstract and title. Further, 93 studies were subjected to eligibility assessment, among which 52 of them were removed due to the presence of unrepresentative sample, redundancy of reports and the published outcome variables are different from the inclusion criteria of this study. Of the screened studies for eligibility, only 46 reports met the inclusion criteria for the systematic review and meta-analysis (Fig. 1). The 46 full-length studies that fulfilled the inclusion criteria were subjected to the analysis of the pooled prevalence of bovine mastitis and the corresponding bacterial causative agents.

## Study characteristics

The characteristics of the selected studies for the meta-analysis are described (Table 1). All the retrieved studies were written in English. As shown in Table 1, a total of 46 studies
**Table 1 Characteristics of studies describing the prevalence of bovine bacterial mastitis.**

| Author, publication year | Study year | Cases | Sample size | APP | Diagnostic method | QS | Production system | Study site | Region | Total isolate | Total sample |
|---|---|---|---|---|---|---|---|---|---|---|---|
| Abebe et al., 2016 | 2015 | 331 | 529 | 0.592 | CE, CMT, Bac | 5 | Intensive | Hawassa | SNNP | 88 | 529 |
| Abebe et al., 2020 | 2018 | 372 | 686 | 0.481 | CE, CMT, Bac | 6 | Intensive | Hawassa | SNNP | 310 | 686 |
| Abera et al., 2012 | 2009 | 61 | 201 | 0.254 | CE, CMT, Bac | 6 | Intensive | Hawassa | SNNP | 70 | 201 |
| Abunna et al., 2013 | 2009 | 173 | 331 | 0.368 | CE, CMT, Bac | 8 | Semi-intensive | Addis Ababa | Addis Ababa | 71 | 331 |
| Adane et al., 2017 | 2012 | 35 | 384 | 0.073 | CE, CMT, Bac | 5 | Semi-intensive | Jigiga | Somali | 35 | 384 |
| Alagaw et al., 2017 | 2015 | 84 | 320 | 0.225 | CE, CMT, Bac | 5 | Semi-intensive | Wolaita Sodo | SNNP | 126 | 320 |
| Amdhun et al., 2016 | 2010 | 62 | 384 | 0.109 | CE, CMT, Bac | 5 | Extensive | Tullo District | Oromia | 119 | 384 |
| Amin et al., 2017 | 2016 | 189 | 384 | 0.417 | CE, CMT, Bac | 5 | Intensive | Haramaya Town | Oromia | 49 | 384 |
| Belina et al., 2016 | 2014 | 237 | 471 | 0.407 | CE, CMT, Bac | 5 | Semi-intensive | Borana | Oromia | 47 | 471 |
| Birhanu et al., 2017 | 2016 | 170 | 262 | 0.161 | CMT, Bac | 8 | Intensive | Debre Zeit | Oromia | 159 | 262 |
| Bitew et al., 2010 | 2009 | 85 | 302 | 0.252 | CE, CMT, Bac | 5 | Semi-intensive | Bahir Dar | Amhara | 76 | 302 |
| Bogale et al., 2018 | 2010 | 393 | 1,019 | 0.339 | CE, CMT, Bac | 6 | Extensive | West Hararghe | Oromia | 393 | 1,019 |
| Demissie et al., 2018 | 2016 | 130 | 360 | 0.267 | CE, CMT, Bac | 5 | Intensive | Wukro | Tigray | 159 | 360 |
| Dereje et al., 2018 | 2015 | 131 | 186 | 0.650 | CE, CMT, Bac | 5 | Intensive | Holeta | Oromia | 87 | 186 |
| Duguma et al., 2013 | 2010 | 73 | 90 | 0.733 | CE, CMT, Bac | 5 | Intensive | Holeta | Oromia | 155 | 360 |
| Disasa et al., 2016 | 2015 | 131 | 334 | 0.332 | CE, CMT, Bac | 5 | Intensive | Dire Dawa | Dire Dawa | 96 | 334 |
| Elemo et al., 2018 | 2014 | 141 | 384 | 0.317 | CE, CMT, Bac | 5 | Semi-intensive | Sinana | Oromia | 406 | 1,536 |
| Gebisa et al., 2019 | 2019 | 194 | 474 | 40.92 | CE, CMT, Bac | 7 | Semi-intensive | Buno Bedele | Oromia | 106 | 474 |
| Getahun et al., 2007 | 2007 | 168 | 500 | 0.304 | CE, CMT, Bac | 9 | Semi-intensive | Selalle | Oromia | 195 | 500 |
| Haftu et al., 2012 | 2010 | 114 | 305 | 0.338 | CE, CMT, Bac | 8 | Intensive | Mekelle | Tigray | 111 | 305 |
| Hailemeskel et al., 2014 | 2012 | 128 | 144 | 0.806 | CE, CMT, Bac | 6 | Semi-intensive | North Showa | Amhara | 41 | 144 |
| Kedir et al., 2016 | 2015 | 131 | 334 | 0.332 | CE, CMT, Bac | 5 | Semi-intensive | Dire Dawa | Dire Dawa | 130 | 334 |
| Kitila et al., 2021 | 2017 | 210 | 532 | 0.227 | CE, CMT, Bac | 5 | Extensive | West Wollega | Oromia | 129 | 532 |
| Kumbe et al., 2020 | 2018 | 187 | 384 | 0.354 | CE, CMT, Bac | 5 | Extensive | Borena zone | Oromia | 133 | 384 |
| Lakew et al., 2009 | 2008 | 144 | 223 | 0.381 | CE, CMT, Bac | 9 | Intensive | Asella | Oromia | 117 | 223 |
| Lakew et al., 2019 | 2019 | 242 | 384 | 0.562 | CE, CMT, Bac | 8 | Intensive | Haramaya District | Oromia | 25 | 384 |
| Madalcho, 2019 | 2012 | 124 | 349 | 0.301 | CE, CMT, Bac | 5 | Semi-intensive | Wolaita Sodo | SNNP | 111 | 349 |
| Mekibib et al., 2010 | 2009 | 76 | 107 | 0.486 | CE, CMT, Bac | 6 | Intensive | Holeta Town | Oromia | 153 | 428 |
| Mekonen et al., 2017 | 2016 | 316 | 510 | 0.62 | CE, CMT, Bac | 7 | Intensive | Bahir Da | Amhara | 149 | 510 |
| Mekonnen & Tesfaye, 2010 | 2010 | 99 | 206 | 0.417 | CE, CMT, Bac | 8 | Intensive | Adama | Oromia | 88 | 206 |
| Meranga, 2012 | 2012 | 81 | 111 | 0.568 | CE, CMT, Bac | 6 | Intensive | Alage State | Oromia | 138 | 441 |
| Nahom, 2021 | 2020 | 72 | 422 | 0.152 | CE, CMT, Bac | 7 | Semi-intensive | Gamo Zone | SNNP | 64 | 422 |
| Pal et al., 2017 | 2013 | 100 | 195 | 0.512 | CMT, Bac | 5 | Intensive | Asella | Oromia | 126 | 195 |
| Seid et al., 2015 | 2014 | 136 | 358 | 0.307 | CE, CMT, Bac | 5 | Semi-intensive | West Arsi | Oromia | 80 | 358 |
| Shiferaw & Telila, 2016 | 2014 | 134 | 386 | 0.212 | CE, CMT, Bac | 5 | Semi-intensive | Wolaita Sodo | SNNP | 74 | 386 |
| Tadesse & Chanie, 2012 | 2011 | 196 | 300 | 0.433 | CE, CMT, Bac | 6 | Intensive | Addis Ababa | Addis Ababa | 93 | 300 |
| Tegegne et al., 2020 | 2016 | 214 | 303 | 0.706 | CE, CMT, Bac | 5 | Intensive | Holeta | Oromia | 187 | 303 |

| Author, publication year | Study year | Cases | Sample size | APP | Diagnostic method | QS | Production system | Study site | Region | Total isolate | Total sample |
|---|---|---|---|---|---|---|---|---|---|---|---|
| Tekle & Berhie, 2016 | 2011 | 43 | 96 | 0.427 | CE, CMT, Bac | 5 | Semi-intensive | Sidama | SNNP | 39 | 96 |
| Teklesilasie et al., 2014 | 2012 | 192 | 365 | 0.449 | CE, CMT, Bac | 6 | Intensive | Addis Ababa | Addis Ababa | 121 | 365 |
| Tesfaye & Abera, 2018 | 2017 | 136 | 216 | 0.606 | CE, CMT, Bac | 5 | Intensive | Jimma | SNNP | 323 | 841 |
| Tesfaye, 2017 | 2017 | 78 | 351 | 0.207 | CE, CMT, Bac | 5 | Semi-intensive | Addis Ababa | Addis Ababa | 198 | 351 |
| Workineh et al., 2002 | 1998 | 111 | 186 | 0.382 | CE, CMT, Bac | 9 | Intensive | Debre Zeit | Oromia | 121 | 186 |
| Yenew & Addis, 2020 | 2017 | 49 | 180 | 0.239 | CE, CMT, Bac | 7 | Semi-intensive | Dessie | Amhara | 76 | 180 |
| Yohannis & Mola, 2013 | 2012 | 103 | 349 | 0.269 | CE, CMT, Bac | 5 | Semi-intensive | Wolaita | SNNP | 90 | 349 |
| Zeryehun & Abera, 2017 | 2016 | 247 | 384 | 0.518 | CE, CMT, Bac | 8 | Intensive | Eastern Hararghe | Oromia | 187 | 384 |
| Zeryehun et al., 2013 | 2013 | 373 | 499 | 0.551 | CE, CMT, Bac | 5 | Intensive | Addis Ababa | Addis Ababa | 80 | 499 |

with 7,196 cow-level cases and 5,931 bacterial isolates were included for systematic review and meta-analysis. A total of 15,780 cows and 18,478 suspected milk samples were enrolled in the studies, among which 7,196/15,780 (45.6%) were identified to be in diseased category and they are infected by any of the 13 reported infectious agents (annexed in the Supplemental Materials). The mean sample size of the 46 studies was 343 ± 165.4 cows. Among the 46 studies, three studies focused on estimating the prevalence and isolation of pathogens related to subclinical mastitis whereas the remaining studies included both types of bovine mastitis (*Pal, Lemu & Bilata, 2017*; *Mekonnen et al., 2017*; *Tegegne et al., 2020*). Except for two studies, all studies performed a physical examination of the mammary gland and subsequent collection of milk samples for analysis. Likewise, they have performed the California mastitis test and applied standard microbiological culturing techniques for the isolation of potential bacterial agents (*Pal, Lemu & Bilata, 2017*) (*Birhanu et al., 2017*). Concerning the regional states of study in the Ethiopia, 21 studies were from Oromia, 10 studies from SNNPR, five studies from Addis Ababa, four studies from Amhara, two studies from Somali, two studies from Dire Dawa and two studies from Tigray regional states (Table 1).

## Geographical distribution of studies in Ethiopia

The retrieved studies were conducted between 1995 and 2021 in seven regional states, namely Amhara, Oromia, Gambella, Southern Nations Nationalities and Peoples Region (SNNPR), Somali, Tigray and in the two city administrative areas such as Addis Ababa and Dire Dawa. The geographic distribution of the cases in various regional states of Ethiopia indicates that 21 studies were from the Oromia region, 10 studies from SNNPR, five studies from Addis Ababa, four studies from Amhara and two studies each from Somali, Dire Dawa and Tigray regional states respectively (Table 1). The spatial distribution of studies by location was described in Fig. 2. The pooled apparent prevalence
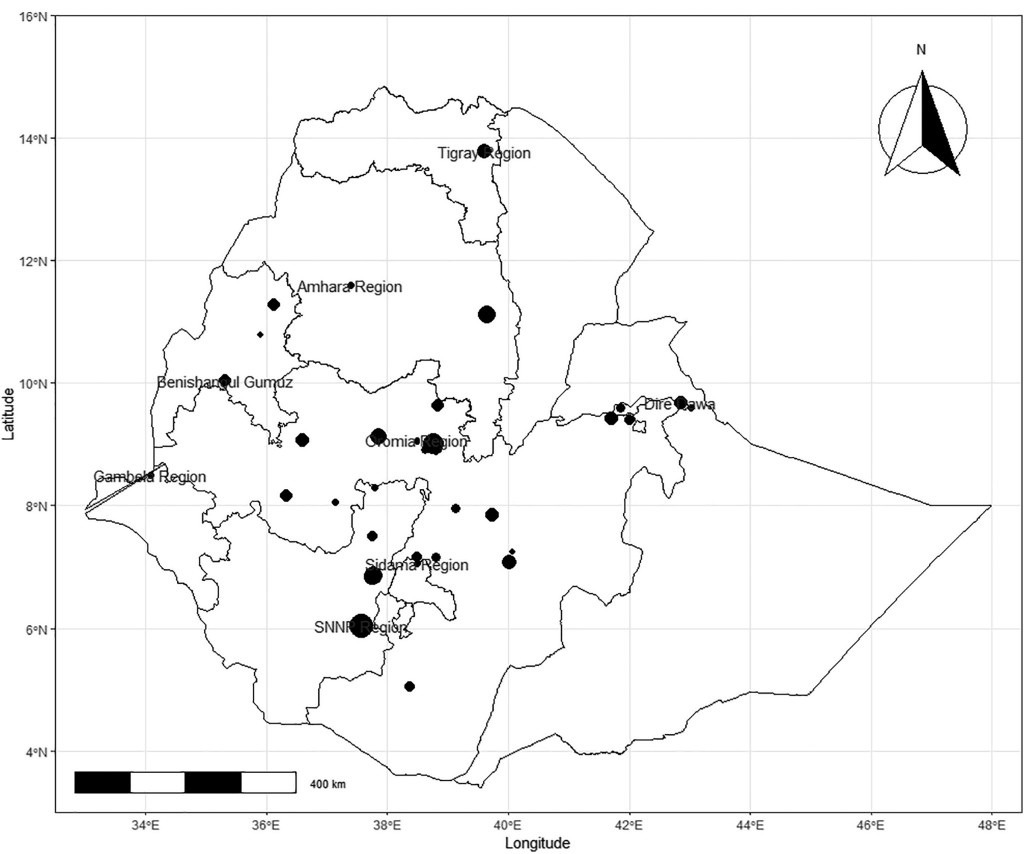

**Figure 2 Observed spatial distribution of bacterial bovine mastitis in Ethiopia.**

of regional states were 38.2%, 30.9%, 35.8%, 30.2%, 33.6%, 40.2%, and 19.6% in Oromia, Amhara, SNNPR, Addis Ababa, Dire Dawa, Tigray and Somali, respectively (Fig. 2).

## Meta-analysis and meta-regression

The meta-analysis of the pooled prevalence of mastitis was conducted by using data of 15,780 cows from 46 studies. The pooled estimate of the overall prevalence of bovine mastitis based on random effect model was 47.6% (95% CI [42.4–52.9]). The heterogeneity of this study was found to be significant ($Q = 2,020$, df = 45, $P < 0.0001$) between 46 studies. The value of inverse variance index ($I^2$) was predicted as 98% and it indicates the presence of high variations between studies. Further, the factors that could explain this variance were tested by subsequent meta-regression approach. Applying the one study removal method showed that omitting any of the studies did not alter the pooled overall prevalence in a significant manner, $P > 0.05$. Therefore, it signifies that the single study omitted estimates lie within the 95% CI of the overall mean.

### Pooled prevalence

The pooled prevalence estimate for the mastitis occurrence among dairy cows by the random effect model was 47.6% with 95% CI [42.4–52.9] as shown in the forest plot

(Fig. 3). The Cochran's $Q$ value of 2,020.6 (df = 45, $P < 0.001$) indicates significant variation in true effects. The proportion of variance between the studies was high as indicated by the $\tau^2$ estimate of 0.0321 and heterogeneity ($I^2$) value of 98.4%.

### Prevalence of bacterial mastitis

The pooled estimate of bacterial mastitis from the isolates was 33.1% (95% CI [29.1–37.3]) with a degree of heterogeneity ($I^2$) value of 97% (Fig. 4). The univariate meta-regression was not shown significant association for the variables like year of study, size of the sample, production system, regions in the country, and diagnostic method; however, significance attained between the type of mastitis and system of production ($P$ value < 0.05) (Table 2). The subgroup analysis and sensitivity testing was performed based on the type of mastitis and production system (Table 3). The rationale for this analysis is to observe the change in the degree of heterogeneity excluding the known outliers.

### Prevalence of bacterial isolates

The subgroup analysis has shown that *Staphylococcus aureus* was the major isolate accounting to 13.4% and 16.5% for the clinical and subclinical mastitis, respectively followed by *Streptococci (Strep)* species, with the prevalence of 10.6% and 8.1% for clinical and subclinical mastitis, respectively (Table 3). The proportion of highest bacterial isolates was obtained among cows inflicted by subclinical mastitis (33.3%) with 98% degree of heterogeneity (Table 3). Among the cows infected by mastitis and those under intensive system, the prevalence of bacterial mastitis was 39.5% ($I^2 = 97$%) (Table 4). The other main bacterial isolates were *Strep agalactia* and *Strep. dysagalactia, Escherchia coli* (*E. coli*), *Klebsiella* species, Coagulase Negative *Staphylococci* (CoNS) and *Corynebacterium* species (mainly *Coryne. bovis*), *Bacillus* species, *Micrococcus* species and *Enterobacter* species (Tables 3 and 4).

On the other hand, the subgroup analysis of production system for bacteria as a cause of mastitis has shown certain species of pathogenic bacteria like *Strep. dysagalactia, Strep. uberis* and *Bacillus* species have not isolated under the extensive system managed cows (Table 4). Despite the *E. coli* and *Streptococcus* species were found to have a relatively higher prevalence in cows managed under intensive production, the prevalence of *Staph. aureus* and CoNS remained high under intensive production with the predicted values of 15.5% and 10.1%, respectively. However, when combined together with other pathogens it accounted for 39% of bacterial mastitis (Table 4).

### Temporal distribution of the studies

The distribution of studies retrieved from 1998 to 2021 indicated the temporal pattern across the study years. Of the all retrieved studies, 74% meant the majority of studies got published only after the year 2010 (Fig. 5). We observed a paucity of research work between the years 1998 to 2008 (6.5%, *i.e.* 3/46 studies). The time series model investigation to predict the trend of the number of studies over time showed the decreasing starting from the year 2015 with a trend line of equation $Y = -0.047X + 94.2$. The intercept value for the aforementioned equation was indicated a decreased undergoing of studies

| Study | Events | Total | Events per 100 observations | Events | 95%-CI | Weight (common) | Weight (random) |
|---|---|---|---|---|---|---|---|
| Abebe et al., 2016 | 331 | 529 | | 62.57 | [58.29; 66.71] | 3.4% | 2.2% |
| Abebe et al., 2020 | 372 | 686 | | 54.23 | [50.41; 58.00] | 4.3% | 2.2% |
| Abera et al., 2012 | 61 | 201 | | 30.35 | [24.08; 37.21] | 1.3% | 2.2% |
| Abunna et al., 2013 | 173 | 331 | | 52.27 | [46.73; 57.76] | 2.1% | 2.2% |
| Adane et al., 2017 | 35 | 384 | | 9.11 | [6.43; 12.45] | 2.4% | 2.2% |
| Alagaw et al., 2017 | 84 | 320 | | 26.25 | [21.51; 31.43] | 2.0% | 2.2% |
| Amdhun et al., 2016 | 62 | 384 | | 16.15 | [12.61; 20.21] | 2.4% | 2.2% |
| Amin et al., 2017 | 189 | 384 | | 49.22 | [44.11; 54.34] | 2.4% | 2.2% |
| Belina et al., 2016 | 237 | 471 | | 50.32 | [45.71; 54.93] | 3.0% | 2.2% |
| Birhanu et al., 2017 | 170 | 262 | | 64.89 | [58.77; 70.66] | 1.7% | 2.2% |
| Bitew et al., 2010 | 85 | 302 | | 28.15 | [23.14; 33.58] | 1.9% | 2.2% |
| Bogale et al., 2018 | 393 | 1019 | | 38.57 | [35.57; 41.63] | 6.5% | 2.2% |
| Demissie et al., 2018 | 130 | 360 | | 36.11 | [31.14; 41.31] | 2.3% | 2.2% |
| Dereje et al., 2018 | 131 | 186 | | 70.43 | [63.31; 76.88] | 1.2% | 2.1% |
| Duguma et al., 2013 | 73 | 90 | | 81.11 | [71.49; 88.59] | 0.6% | 2.1% |
| Disasa et al, 2016 | 131 | 334 | | 39.22 | [33.95; 44.68] | 2.1% | 2.2% |
| Elemo et al., 2018 | 141 | 384 | | 36.72 | [31.89; 41.76] | 2.4% | 2.2% |
| Gebisa et al., 2019 | 194 | 474 | | 40.93 | [36.46; 45.51] | 3.0% | 2.2% |
| Getahun et al., 2007 | 168 | 500 | | 33.60 | [29.47; 37.93] | 3.2% | 2.2% |
| Haftu et al., 2012 | 114 | 305 | | 37.38 | [31.93; 43.07] | 1.9% | 2.2% |
| Hailemeskel et al., 2014 | 128 | 144 | | 88.89 | [82.58; 93.51] | 0.9% | 2.1% |
| Kedir et al., 2016 | 131 | 334 | | 39.22 | [33.95; 44.68] | 2.1% | 2.2% |
| Kitila et al., 2021 | 210 | 532 | | 39.47 | [35.29; 43.77] | 3.4% | 2.2% |
| Kumbe et al., 2020 | 187 | 384 | | 48.70 | [43.60; 53.82] | 2.4% | 2.2% |
| Lakew et al., 2009 | 144 | 223 | | 64.57 | [57.91; 70.84] | 1.4% | 2.2% |
| Lakew et al., 2019 | 242 | 384 | | 63.02 | [57.98; 67.86] | 2.4% | 2.2% |
| Madalcho, 2019 | 124 | 349 | | 35.53 | [30.51; 40.80] | 2.2% | 2.2% |
| Mekibib et al., 2010 | 76 | 107 | | 71.03 | [61.46; 79.39] | 0.7% | 2.1% |
| Mekonen et al, 2017 | 316 | 510 | | 61.96 | [57.59; 66.19] | 3.2% | 2.2% |
| Mekonnen and Tesfaye, 2010 | 99 | 206 | | 48.06 | [41.06; 55.11] | 1.3% | 2.2% |
| Meranga, 2012 | 81 | 111 | | 72.97 | [63.72; 80.96] | 0.7% | 2.1% |
| Nahom, 2021 | 72 | 422 | | 17.06 | [13.60; 21.00] | 2.7% | 2.2% |
| Pal et al., 2017 | 100 | 195 | | 51.28 | [44.04; 58.49] | 1.2% | 2.2% |
| Seid et al., 2015 | 136 | 358 | | 37.99 | [32.94; 43.24] | 2.3% | 2.2% |
| Shiferaw and Telila, 2016 | 134 | 386 | | 34.72 | [29.97; 39.70] | 2.4% | 2.2% |
| Tadesse and Chanie, 2012 | 196 | 300 | | 65.33 | [59.65; 70.71] | 1.9% | 2.2% |
| Tegegne et al., 2020 | 214 | 303 | | 70.63 | [65.15; 75.70] | 1.9% | 2.2% |
| Tekle and Berhie, 2016 | 43 | 96 | | 44.79 | [34.63; 55.29] | 0.6% | 2.1% |
| Teklesilasie et al., 2014 | 192 | 365 | | 52.60 | [47.34; 57.82] | 2.3% | 2.2% |
| Tesfaye and Abera, 2018 | 136 | 216 | | 62.96 | [56.15; 69.42] | 1.4% | 2.2% |
| Tesfaye, 2017 | 78 | 351 | | 22.22 | [17.98; 26.94] | 2.2% | 2.2% |
| Workineh et al., 2002 | 111 | 186 | | 59.68 | [52.25; 66.79] | 1.2% | 2.1% |
| Yenew and Addis, 2020 | 49 | 180 | | 27.22 | [20.87; 34.34] | 1.1% | 2.1% |
| Yohannis and Mola, 2013 | 103 | 349 | | 29.51 | [24.78; 34.60] | 2.2% | 2.2% |
| Zeryehun and Abera, 2017 | 247 | 384 | | 64.32 | [59.31; 69.12] | 2.4% | 2.2% |
| Zeryehun et al., 2013 | 373 | 499 | | 74.75 | [70.70; 78.51] | 3.2% | 2.2% |
| **Common effect model** | | 15780 | | 45.28 | [44.51; 46.06] | 100.0% | -- |
| **Random effects model** | | | | 47.65 | [42.41; 52.92] | -- | 100.0% |

Heterogeneity: $I^2 = 98\%$, $\chi^2_{45} = 2020.58$ ($p = 0$)

20 40 60 80

**Figure 3 Pooled prevalence of bovine mastitis occurrence in Ethiopia.** Forest plot depicting the pooled prevalence of bovine mastitis occurrence in Ethiopia.

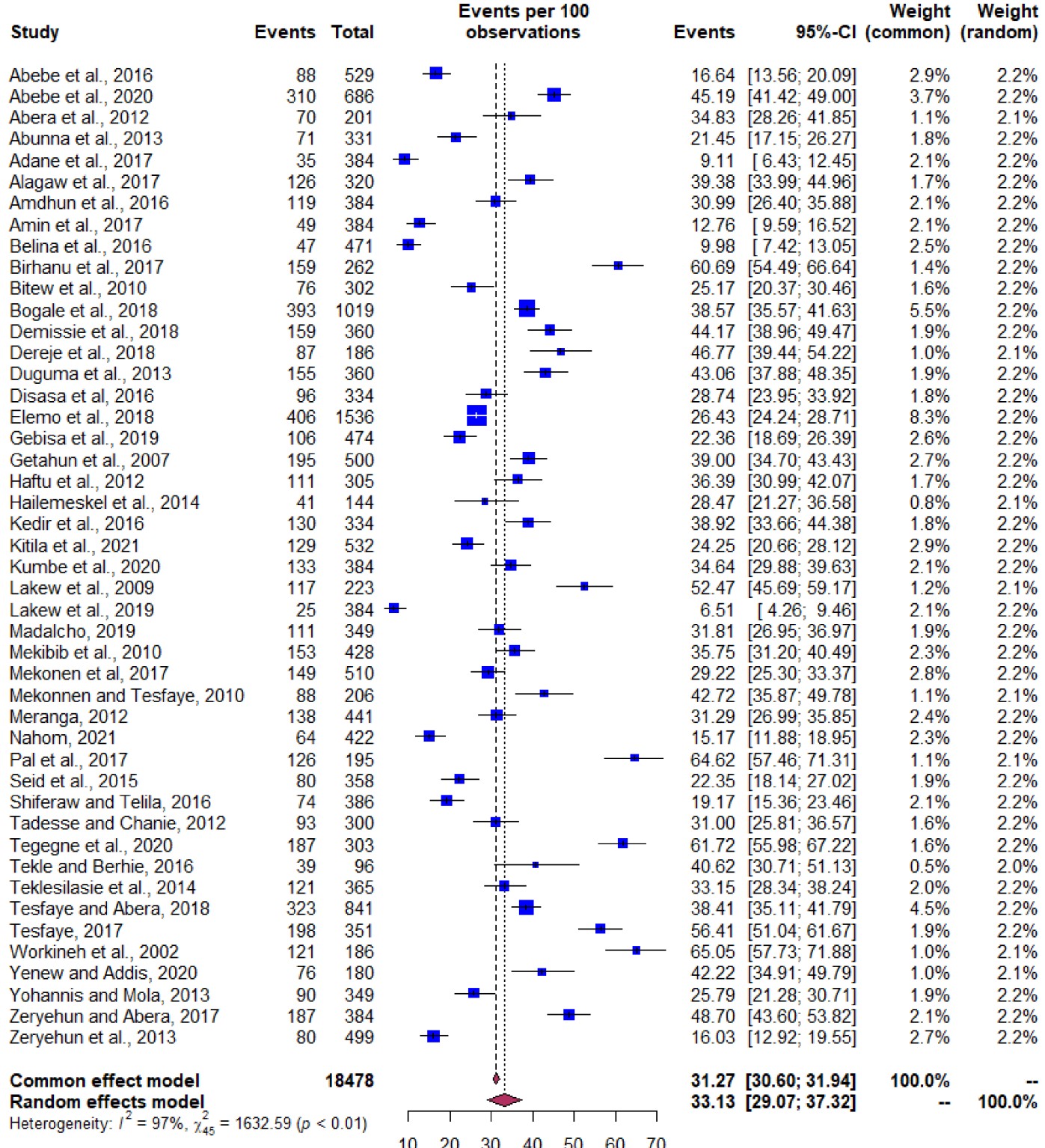

**Figure 4 Forest plot depicting pooled prevalence of bacterial mastitis in Ethiopia.**

**Table 2 Pooled estimates of mastitis prevalence by potential moderators.**

| Risk factor | No. of studies | Prev (95% CI) | I² (%) | χ² | OR (95% CI) | P-value |
|---|---|---|---|---|---|---|
| Study year | | | | | | |
| 2010–2021 | 34 | 35.1 [26.8–44.1] | 98 | 1,410 | Reference | |
| 1995–2010 | 12 | 43.3 [26.6–59.4] | 99 | 734.4 | 1.43 [0.82–2.5] | 0.2145 |
| Production system | | | | | | |
| Extensive | 4 | 25.5 [8.1–47.6] | 99 | 642.8 | Reference | |
| Semi-intensive | 18 | 28.2 [21.7–39.5] | 97 | 558.4 | 1.54 [1.22–1.42] | 0.0481 |
| Intensive | 24 | 36.5 [26.3–45.8] | 97 | 860.7 | 1.75 [1.31–2.85] | 0.0321 |
| Type of mastitis | | | | | | |
| Clinical | 24 | 6.56 [4.24–10.3] | 97 | 941.7 | Reference | |
| Subclinical | 46 | 28.3 [21.4–36.3] | 95 | 2,360 | 1.2 [1.14 –2.12] | <0.001 |
| Diagnostic method | | | | | | |
| CE, CMT, Bac | 44 | 31 [23.7–39.9] | 98 | 2,172 | Reference | |
| CMT, Bac | 2 | 54.6 [21.5–78.8] | 0 | 0.73 | 2.96 [0.88–10.2] | 0.0773 |
| Regions in Ethiopia | | | | | | |
| Oromia | 22 | 42.6 [31.3-54.7] | 99 | 1,520 | Reference | |
| Amhara | 4 | 30.9 [12.5–58.5] | 81 | 16.11 | 1.22 [0.32–3.34] | 0.9531 |
| SNNPR | 10 | 35.8 [21.3–53.6] | 98 | 367.8 | 1.28 [0.49–3.54] | 0.6110 |
| Somali | 1 | 9.1 [0.99–50.08] | 0 | 0 | 0.23 [0.03–1.61] | 0.1392 |
| Tigray | 2 | 40.2 [11.8–77.2] | 76 | 4.13 | 1.56 [0.34–6.74] | 0.5543 |
| Addis Ababa | 5 | 30.2 [13.4–54.6] | 97 | 158.7 | 1.28 [0.49–3.32] | 0.6110 |
| Dire Dawa | 2 | 33.6 [9.2–71.8) | 87 | 7.69 | 1.17 [0.27–5.08] | 0.8314 |
| Sample size | | | | | | |
| <384 | 33 | 40.7 [31.8–50.2] | 97 | 1,072 | Reference | |
| ≥384 | 13 | 28.6 [17.9–42.5] | 99 | 1,180 | 0.58 [0.32–1.04] | 0.0693 |

(−0.047) that provided a considerable variation existence among the number of studies over time (Fig. 5).

## Publication bias

We assessed the publication bias and small study effects by funnel plot observation, Begg's and Egger's test. Publication bias is actually just one of many reporting biases that could distort the evidence that we obtain in our meta-analysis. These might include citation bias, time-lag bias, multiple publication bias, language bias and outcome reporting bias. The evidence of publication bias was identified from the funnel plot, which graphically represent the data obtained from the prevalence of bacterial isolates and includes the standard error with augmented proportion (Figs. 6A and 6B). The Egger test revealed that the funnel plot was significantly asymmetrical ($z = 3.3537$, $p = 0.0008$) (95% CI = [−1.49 to −0.59]; $P$ value of 0.012) and confirms the presence of publication bias. Moreover, we could also observe the theoretical missing of study when we incorporated by Duval and Tweedie's trim and fill method. Based on the aforementioned evidences of publication bias, RE model would be more appropriate for this data.

**Table 3 Subgroup analysis of bacterial isolates based on the type of bovine mastitis.**

| Bacterial isolate | Type of mastitis, prevalence % (95% CI) | | | | | |
| --- | --- | --- | --- | --- | --- | --- |
| | Cows with clinical mastitis | | | Cows with subclinical mastitis | | |
| | Pooled prevalence (%) | No. of studies | $I^2$ (%) | Pooled prevalence (%) | No. of studies | $I^2$ (%) |
| Pooled isolates per | 6.5 [4.2–10.3] | 24 | 95 | 28.3 [21.1–36.3] | 46 | 98 |
| *Staph. Aureus* | 16.7 [11.7–23.2] | 11 | 92 | 13.4 [10.9–16.3] | 34 | 94 |
| *Streptococcus* spp. | 10.6 [6.5–17.1] | 10 | 99 | 8.1 [5.9–10.7] | 29 | 92 |
| *Strep. agalactia* | 5.7 [3.3–9.6] | 7 | 96 | 4.8 [3.4–6.7] | 19 | 90 |
| *Strep. dysagalactia* | 3.1 [1.6–5.7] | 4 | 66 | 2.8 [1.9–4.1] | 13 | 76 |
| *Strep. uberis* | 2.7 [1.1–7.2] | 2 | 0 | 2.5 [1.7–3.6] | 14 | 72 |
| *Staph. epidermis* | 9.8 [4.6–19.7] | 2 | 0 | 5.9 [3.8–9.1] | 7 | 93 |
| *Escherchia coli* | 3.1 [1.5–6.3] | 10 | 88 | 3.6 [2.3–5.5] | 26 | 93 |
| CoNS | 8.4 [4.8–13.9] | 5 | 78 | 8.6 [6.8–10.8] | 25 | 90 |
| *Klebsiella* spp. | 2.3 [0.9–5.7] | 4 | 12 | 1.9 [1.1–3.4] | 13 | 94 |
| *Coryn.* spp. | 1.6 [0.3–7.7] | 2 | 13 | 2.1 [1.3–3.3] | 21 | 87 |
| *Enterobacter* spp. | 0.7 [0.14–3.6] | 2 | 69 | 2.9 [1.2–7.1] | 6 | 90 |
| *Micrococcus* spp. | 1.2 [0.3–4.9] | 3 | 89 | 2.1 [1.2–3.5] | 18 | 81 |
| *Bacillus* spp. | 3.2 [1.2–8.2] | 3 | 14 | 1.5 [1.0–2.3] | 17 | 72 |

**Table 4 Subgroup analysis of bacterial isolates based on the type of production system.**

| Bacterial isolates | Types of production system, prevalence % (95% CI) | | | | | | | | |
| --- | --- | --- | --- | --- | --- | --- | --- | --- | --- |
| | Extensive production | | | Intensive production | | | Semi-extensive production | | |
| | Pooled prevalence (%) | No. of studies | $I^2$ (%) | Pooled prevalence (%) | No. of studies | $I^2$ (%) | Pooled prevalence (%) | No. of studies | $I^2$ (%) |
| Pooled isolates per | 25.5 [8.1–47.6] | 4 | 99 | 36.5 [26.3–45.8] | 24 | 97 | 28 [21.7–39.5] | 18 | 97 |
| *Staph. Aureus* | 14.23 [4.5–36.5] | 4 | 67 | 15.5 [9.7–22.3] | 22 | 94 | 12.3 [7.3–19.7] | 19 | 95 |
| *Streptococcus species* | 18.7 [8.2–37.4] | 3 | 99 | 7.8 [5.7–10.8] | 23 | 88 | 8.4 [5.4–12.7] | 13 | 94 |
| *Strep. agalactia* | 7.9 [2.91–19.7] | 3 | 98 | 5.1 [3.4–7.5] | 14 | 60 | 4.5 [2.8–7.2] | 10 | 94 |
| *Strep. dysagalactia* | – | – | – | 3.2 [2.12–4.86] | 9 | 73 | 2.5 [1.62–3.9] | 8 | 75 |
| *Strep. uberis* | – | – | – | 2.7 [1.7–4.3] | 9 | 76 | 2.4 [1.3–3.8] | 7 | 55 |
| *Staph. epidermis* | 8.2 [3.7–17.1] | 3 | 96 | 4.3 [1.8–9.5] | 3 | 76 | 7.3 [4.4–12.1] | 5 | 67 |
| *Escherchia coli* | 10.1 [4.3–36.2] | 2 | 58 | 5.1 [3.1–7.5] | 19 | 86 | 1.7 [1.1–2.9] | 15 | 92 |
| CoNS | 7.8 [2.5–21.7] | 1 | – | 10.1 [7.8–20.8] | 18 | 88 | 6.7 [4.6–9.3] | 11 | 90 |
| *Klebsiella* | 7.8 [1.4–32.9] | 1 | – | 10 [6.82–14.4] | 18 | 88 | 6.4 [3.8–10.4] | 11 | 90 |
| *Corynebacterium* spp. | 6.5 [4.12–30.1] | 1 | – | 2.3 [1.3–4.12] | 12 | 90 | 2.2 [0.7–3.1] | 10 | 78 |
| *Enterobacter* spp. | 2.2 [0.6–5.8] | 5 | 84 | 1.8 [0.4–8.3] | 3 | 91 | – | – | – |
| *Micrococcus* spp. | 1.4 [0.5-4.1] | 4 | 75 | 3.5 [1.9–6.5] | 9 | 81 | 1.08 [0.5–2.3] | 8 | 72 |
| *Bacillus* spp. | – | – | – | 2.6 [1.76–3.9] | 11 | 66 | 1.1 [0.6–1.72] | 9 | 25 |

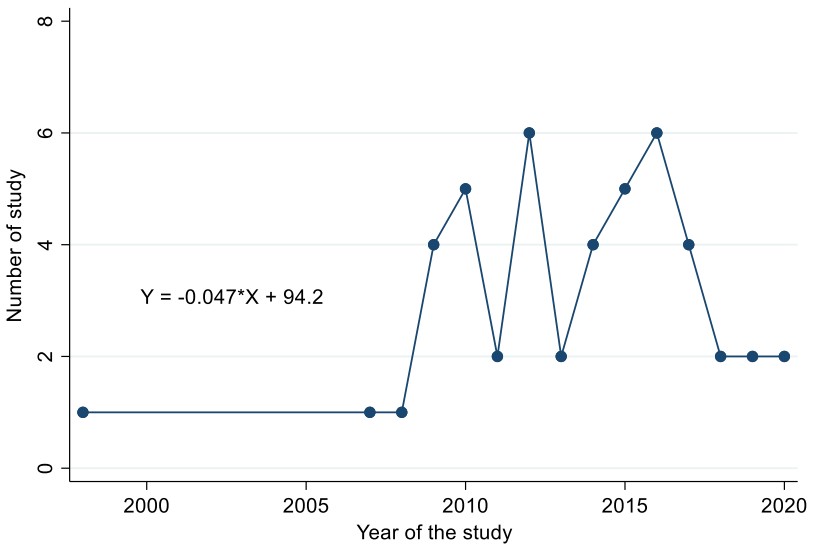

**Figure 5 Trend line and equation showing growing trend of published articles over years.**

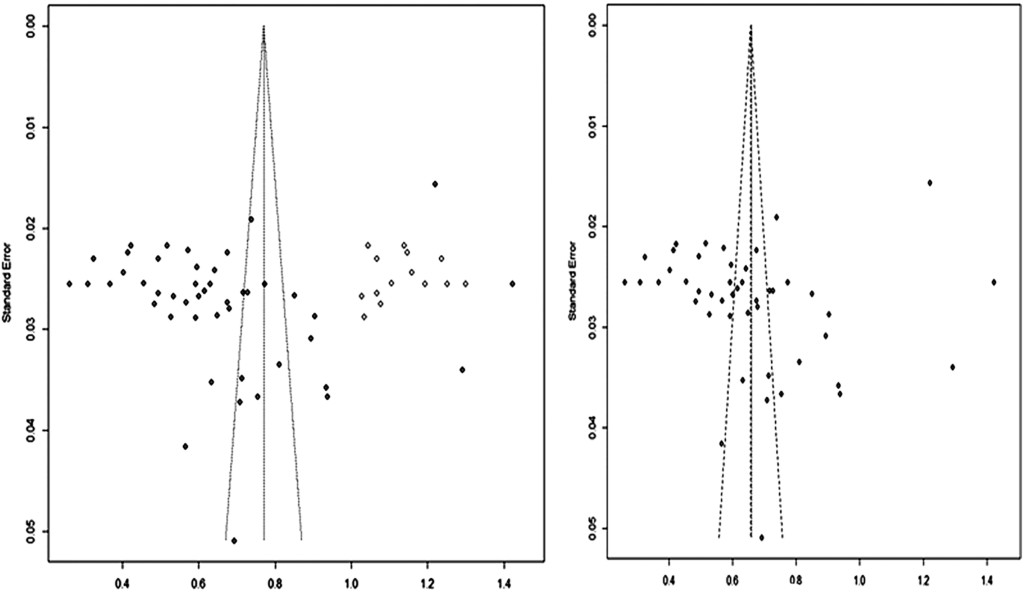

**Figure 6 Funnel plots indicating the presence of publication bias.** Freeman-Tukey double arcsine transformed proportion.

## DISCUSSION

To the best of our knowledge, this is the first systematic review and meta-analysis conducted on the prevalence of bacterial mastitis in dairy cows in Ethiopia. This study recorded the pooled prevalence estimates, spatial and temporal distributions of mastitis in Ethiopia. The analysis included 46 original studies that address the prevalence of bovine mastitis and the associated pathogenic bacterial isolates within the specified time frame.

The suspected milk samples were subjected to microbiological analysis. In all the studies of the isolation of causative bacteria, clinical examination of mammary gland and the California mastitis test were conducted. The majority of studies were reported from the Oromia region (46.5%), while the remaining studies originated from other regions of the country.

Mastitis is one of the predominant and multifactorial cattle disease for which different reports are released from different production systems in Ethiopia. The overall pooled prevalence estimate of bovine mastitis in dairy cows was found to be 47.6% (95% CI [42.4–52.9]). The level of heterogeneity between the studies as predicted by the inverse variance square ($I^2$ = 98%) indicates the greater degree of variation between studies (Fig. 6). In order to restrict the level of heterogeneity between the studies, we attempted to include studies with similar diagnostic method and study design. The possible sources for observed variation could be due to the differences in the agro-climatic conditions, farm management practices and cow associated factors such as genetic makeup of cows, parity, and lactation stages (*Getaneh & Gebremedhin, 2017*).

The pooled estimates of bovine mastitis based on cow associated factors were determined as 47.6%. This estimate is in corroboration with the study conducted by *Getaneh & Gebremedhin (2017)* who reported the 47% prevalence (*n* = 11,669) for the cow's positive for bovine mastitis from January 2002 to June 2016. *Bangar et al. (2015)* also reported that prevalence of mastitis was 46.3% on a cow level studies in India. Moreover, the pooled estimate observed in the present meta-analysis is in agreement of results reported by *Mekonnen & Tesfaye (2010)* but lower than studies (*Abebe et al., 2016*; *Abunna et al., 2013*). In most cases, it was difficult for us to compare the present study estimates with the reports from single studies having an almost similar farming system. Likewise, the scarcity of nationwide survey reports also present as a limiting factor for comparison of our result.

The pooled estimate of bacterial mastitis was found to be 33.1% (95% CI [29.1–37.3]). This high prevalence of bacterial mastitis in Ethiopia may be attributed to the low level of biosecurity, low attention to dry cow therapy and vaccination coverage, poor management practices, and weakened veterinary interventions across the country, particularly in the extensive production system. Of the type of mastitis, subclinical mastitis was reported to be the most prevalent dairy cows' constraints in Ethiopia, followed by calf morbidity and mortality in tropics including the country (*Woods, 1987*; *Moran, 2011*; *Tora et al., 2021*).

The pooled prevalence of subclinical mastitis (SCM) based on bacterial isolation was 28.3% which is significantly higher than clinical mastitis, 6.5%. The high prevalence of SCM may be associated with the lack of proper mastitis management practices.
The setback in the implementation of five-point mastitis control plan in Ethiopia is the potential factors ascribed to increase the prevalence of mastitis. In developed countries, however, the application of mastitis control practices, such as the five-point mastitis control plan has proved to decrease the mastitis prevalence and lower bulk milk somatic cell counts (*Ndahetuye et al., 2019*; *Seegers & Fourichon, 2003*). Low somatic cell count of milk maintains high organoleptic quality for shelf-life period (*Blowey & Edmondson, 2010*). The widespread prevalence of mastitis under intensive management system in

Ethiopia as seen from the meta-analysis results indicate the increase in the presence of infection pressure on the cow, particularly when a mastitis control plan is not implemented. Likewise, the existence of low awareness among the dairy producers and the lack of quality or payment standards for bulk milk somatic cell count in Ethiopia could explain the low stimulus to implement the management practices that prevent and control mastitis. The aforementioned factors are the major reasons for the increase in the SCM prevalence in Ethiopia.

The major factors for the transfer of infection from the reservoir to the teat end are the penetration of the teat canal; dry period and during the lactation infections (*Blowey & Edmondson, 2010*). The contagious pathogens like *Staph aureus*, *Strep agalactiae*, *Strep dysgalactiae*, Coagulase *negative staphylococci* and *Corynebacterium bovis* could be transmitted into udder by penetration of the teat canal during milking were the main bacterial pathogens for mastitis in this analysis. The aforementioned pathogens were included in the meta-analysis and they are the major isolates reported by other published single studies in Ethiopia (*Abebe et al., 2016*; *Belina et al., 2016*; *Boggale et al., 2018*; *Getahun et al., 2008*; *Mekonnen & Tesfaye, 2010*). Other environmental pathogens like *Escherichia coli*, *Klebsiella* species, *Enterobacter*, *Streptococcus uberis* and *Micrococcus* species accounting for 5–9% prevalence could be transmitted between milking and the dry period.

The subgroup analysis of production system for bacteria as a cause of mastitis has shown certain species of pathogenic bacteria like *Staph aureus*, CoNS, *E. coli* and *Strep* species were found to have a higher prevalence 36.5% ($I^2$ = 97%) in cows managed under intensive production. The observed high prevalence in cows kept under intensive production can be related to the breed of cows maintained by this system. The exotic breed of cows was predominantly managed under this system than the indigenous breeds. Many authors have suggested that the pure local breeds are more resistant to contract mastitis than the European breeds (*Lakew, Tolosa & Tigre, 2009*; *Tekle, 2016*). This might be due to the variation in the genetic potential among the breeds that confer disease resistance and adaptation to variable environments. Likewise, the anatomical size of the udder in crossbreed cows is larger, which increase the propensity for contamination and exposure to different pathogens (*Lakew, Fayera & Ali, 2019*; *Mekonnen & Tesfaye, 2010*; *Tekle, 2016*; *Mulunesh Yenew, 2020*).

This meta-analysis study provides updated status about the prevalence of mastitis in dairy cows in Ethiopia and the distribution pattern among various regions of the country. The study highlights the current status of bovine mastitis and the potential causative agents for the disease in Ethiopia. Despite some limitations in the studies included for meta-analysis, the main strength of this review lies in the use of widely accepted methods, inclusive approach in the search process and quality appraisal of the studies (*Munn et al., 2015*). The information presented in this report would help make informed decisions on the control and prevention of bovine mastitis in Ethiopia. The outcomes of these studies would benefit the stakeholders and policymakers to work on the improvement of the dairy industry in Ethiopia.

## CONCLUSION

The current study reports that the pooled estimates of the prevalence of mastitis in Ethiopian dairy cattle were high. The analysis also revealed that *Staphylococcus aureus*, *Streptococcus agalactia Streptococcus dysagalactia*, and Coliforims are the common bacterial pathogens responsible for bovine mastitis in Ethiopia. Subgroup analysis for type of mastitis and the management system was identified as a potential significant factor for bacterial mastitis in Ethiopia. Therefore, the highly prevalent and commonly isolated pathogens which cause contagious mastitis require immediate attention by the government and dairy producers to put them under control by devising robust mastitis prevention and management program. A prompt action is needed to have a strategic implementation of mastitis prevention and vibrant dairy herd health management at least in the major cities and towns of across Ethiopia where more dairy businesses are flourishing.

### Funding

The authors received no funding for this work.

### Competing Interests

The authors declare that they have no competing interests.

### Author Contributions

- Ephrem Toma Tora conceived and designed the experiments, performed the experiments, analyzed the data, prepared figures and/or tables, authored or reviewed drafts of the paper, and approved the final draft.
- Nahom Belay Bekele conceived and designed the experiments, performed the experiments, analyzed the data, authored or reviewed drafts of the paper, and approved the final draft.
- Suresh Kumar R.S. analyzed the data, prepared figures and/or tables, authored or reviewed drafts of the paper, and approved the final draft.

### Data Availability

The raw data is available at Ethiopia - Subnational Administrative Boundaries: https://data.humdata.org/dataset/cod-ab-eth.

### Supplemental Information

Supplemental information for this article can be found online at http://dx.doi.org/10.7717/peerj.13253#supplemental-information.

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
