# Peer review of "Bacterial profile of bovine mastitis in Ethiopia: a systematic review and meta-analysis"

_PeerJ, doi:10.7717/peerj.13253_

## Round 0.1 · original submission · Minor Revisions

Two experts revised this manuscript and found merit enough to be published in this journal after addressing the points raised by the Reviewers.

Reviewer 1 ·

Basic reporting

ok

English needs some work to fix punctuation issues and to check all taxonomic names of all species.

Depict 95% CI consistently in a single format throughout the text.

Figures legends need to be more detailed.

Experimental design

ok

Validity of the findings

ok

Additional comments

Could Figures 4 & 5 be ordered (plotted) by year (decreasing or increasing order)?

I would suggest that the discussion be focused on comparing and contrasting the findings here, without re-instating the results in it.

One point that could enrich the discussion is the potential for bacterial lineage variation across species. For instance, the bacterial population structure can often be stratified at the level of ST or sequence types. Are there any molecular epidemiology data suggesting the spread of single or multiple lineages across the country?

·

Basic reporting

Tora and colleagues present a systematic review of the current literature reporting the prevalence and diagnostics of bovine mastitis in Ethiopia. Also, a meta-analysis was conducted to analyze plausible causes of disease dissemination. I find strengths in this study relevant, for instance, the careful assessment of the literature and the well-designed and conducted statistical analysis. The weakness that this reviewer finds is the writing in some parts that are confusing or contain grammar issues. The findings presented here I agree with the authors that may lead to better policies and redirect resources to tackle the incidence of bovine mastitis.

Experimental design

The criteria used for selecting the reports of bovine mastitis is sound. The statistical analysis is adequate. The only issue this reviewer finds is that the graph in Figure 3 contains a trend line that does not clearly state the purpose. Also, the bias analysis is not clearly described. By going into the raw data, I figured out this, but the conclusions drawn from the data are not clear.

Validity of the findings

The methodology used to generate the review supports the findings reported here, and I believe, makes a solid case for better policymaking. Also, the data is relevant when compared with other countries that take decisive actions against bovine mastitis. In the general comments, I include some specific issues that need attention.

Additional comments

I find that from 2019 to 2022 there are more new reports on bovine mastitis in Ethiopia but as authors state, there is a lacking a comprehensive review summarizing the data from different regions and prevalence.
In the following lines, I provide some suggestions to the manuscript.
Line 9, please correct to “…loss directly attributed to a devastating disease…”
Line 12, please add “has not been established…”
In Line 15, please add bovine mastitis.
Line 18, please replace besides with Additionally,
Line 20, please add “in the English language”
Line 21, please add “…and for the etiology being identified as bacterial mastitis”
Line 30, this reviewer does not fully understand the meaning of this line. Do authors refer to prevalence in the sampled cows? Or they refer to the truly diagnosed cases? Please verify.
In line 31, please write the full names of the species found. In line 36, S. aureus, S. epidermis can be used. Also, “coagulase” should not be in italics.
Line 44 please correct to “a huge national”
Line 59 please add “of dairy cows”
Line 80, please state what an unacceptable age is in this context.
Line 81, please correct “have reported”
Line 86, please correct to as reported, the second ranked…
Line 97, please correct to … bacterial mastitis, the types of mastitis registered and…
In line 107, this reviewer thinks that should be “focused on Ethiopia”
In line 112 please add the second most…
Line 120, please correct to “a comprehensive search of literature reporting bovine…”
In lines 136 o 138 I kindly request to check the writing.
Line 139 I suggest excluding
Line 149, I suggest “are presented in Figure 1”
Line 154, please change to according to these tools (add the reference). This section is a bit confusing; I can understand that the two authors who selected the literature provided a score to each paper and selected those with high scores. The way is written is confusing.
In Figure 1, captions are cut in the lower line, for example, the first box describing the databases used. Please correct this.
The numbering in the result section subheadings is incorrect, please change this (line 203).
Figure 2, the names of the regions are difficult to read, I kindly request that authors improve this figure and include other regions so readers can have a better idea of the distribution in this country.
Figure 3 is not publication grade. I kindly request that the authors improve this by moving the trend line equation so that is not overlapped with data points. The figure legend indicates that the type of trend used and the title in the y axis is not adequate, I suggest a number of studies.
Throughout the results section, please correct the naming of the organisms.
The rationale behind the analysis of Figure 3 as stated in lines 297-302 is not clear.
In the case of publication bias, authors are not indicating what factors are being considered here. Please add this to line 304.
I kindly request an addition to the lines 360-367: I think comparing the data presented here with contrasting data from countries with correct cow and dairy management is important because the policies used differ and the outcome when an outbreak occurs. This suggestion, I think, is relevant for a broader readership.
Also, I suggest (not mandatory but desirable) including a couple of bullet points contrasting the current situation of cow management in Ethiopia with possible solutions. This may strengthen the content of the discussion section and perhaps one day reach policymakers.
In line 405 please modify the “need of hour” statement to a clearer message.
I hope these comments help the authors improve the manuscript; I strongly believe that the publication of this work may help the community of dairy producers in Ethiopia.

---

## Round 0.2 · accepted · Accept

The manuscript was significantly improved.